# Scheduling and Securing Drone Charging System Using Particle Swarm Optimization and Blockchain Technology

**Mohamed Torky** [1,†] , **Mohamed El-Dosuky** [2,3] , **Essam Goda** [4,†] , **Václav Snášel** [5]
**and Aboul Ella Hassanien** [6,*,†]

1 Higher Institute of Computer Science and Information Systems, Culture & Science City, 6 October City 12573, Egypt
2 Faculty of Computers and Information, Mansoura University, Mansoura 35516, Egypt
3 Computer Science Department, Arab East Colleges, Riyadh 53354, Saudi Arabia
4 Arab Academy for Science, Technology and Maritime Transport (AASTMT), Giza 12577, Egypt
5 Faculty of Electrical Engineering and Computer Science, VŠB-Technical University of Ostrava, 70032 Poruba-Ostrava, Czech Republic
6 Faculty of Computers and Artificial Intelligence, Cairo University, Cairo 12613, Egypt
* Correspondence: aboitcairo@cu.edu.eg
† Scientific Research Group in Egypt (SRGE), http://egyptscience.net/ (accessed on 5 February 2021).

**Abstract:** Unmanned aerial vehicles (UAVs) have emerged as a powerful technology for introducing untraditional solutions to many challenges in non-military fields and industrial applications in the next few years. However, the limitations of a drone's battery and the available optimal charging techniques represent a significant challenge in using UAVs on a large scale. This problem means UAVs are unable to fly for a long time; hence, drones' services fail dramatically. Due to this challenge, optimizing the scheduling of drone charging may be an unusual solution to drones' battery problems. Moreover, authenticating drones and verifying their charging transactions with charging stations is an essential associated problem. This paper proposes a scheduling and secure drone charging system in response to these challenges. The proposed system was simulated on a generated dataset consisting of 300 drones and 50 charging station points to evaluate its performance. The optimization of the proposed scheduling methodology was based on the particle swarm optimization (PSO) algorithm and game theory-based auction model. In addition, authenticating and verifying drone charging transactions were executed using a proposed blockchain protocol. The optimization and scheduling results showed the PSO algorithm's efficiency in optimizing drone routes and preventing drone collisions during charging flights with low error rates with an MAE = 0.0017 and an MSE = 0.0159. Moreover, the investigation to authenticate and verify the drone charging transactions showed the efficiency of the proposed blockchain protocol while simulating the proposed system on the Ethereum platform. The obtained results clarified the efficiency of the proposed blockchain protocol in executing drone charging transactions within a short time and low latency within an average of 0.34 s based on blockchain performance metrics. Moreover, the proposed scheduling methodology achieved a 96.8% success rate of drone charging cases, while only 3.2% of drones failed to charge after three scheduling rounds.

**Keywords:** blockchain technology; unmanned aerial vehicles (UAVs); drone scheduling; drone authentication; particle swarm optimization (PSO)

## 1. Introduction

Unmanned aerial vehicles (UAVs) (or drones) are fast becoming an important technology that will play pivotal roles in many non-military fields and industrial applications in the next few years [1,2]. The dependence on drone technology across industries moved quickly from the fad phase to the mega-trend phase as more businesses realized their global usage capabilities, potential, scope, and scale. The global drone market size is forecast to be

worth around USD 102.38 billion by 2030 with expanding growth at a CAGR of 18.2% from 2022 to 2030 [3]. Drone technology can be classified into three usage categories, military, commercial, and personal drones:

Drones will continue to be applied in different military operations due to their high suitability for executing critical and time-sensitive missions and reducing military losses [4]. UAVs have primary roles in military missions [5]. They can be used as target decoys for military tactics and surveillance systems; moreover, they can be used in military research and development. From 2016 to 2021, the demand for military-based drones worldwide reached more than USD 70 billion [6]. The growth of military drone demand is tied to the growth of the border security issues and transnational security threats of countries. The US military remains the biggest market for developing and using military UAVs. Still, countries such as Russia, the UK, China, and Australia have significantly contributed to global military spending on drones.

On the other hand, the commercial usage of UAVs has become the talk of the hour and is gaining an increasing interest in various applications [7]. The commercial UAV industry is still young, but it has seen many integrations and major investments from commercial and industrial conglomerates, chip firms, and IT consulting companies. The special purpose UAV market is expected to grow from USD 9332 million in 2021 to USD 20,548 million in 2026, at a CAGR of 17.1% from 2021 to 2026, and it is predicted to be the biggest market share during the forecast period. It is due to the increasing usage of special purpose UAVs in military and civil operations [6]. It will be great to develop new drone applications in various domains as it becomes cheaper to tailor commercial UAVs. The next generation of commercial drones could soon be performing everyday tasks such as monitoring traffic incidents, fertilizing crop fields and precision crop monitoring, thermal sensor drones for search and rescue operations, surveying hard-to-reach regions, or even delivering fast food meals, etc.

At the level of personal drones, according to Philly By Air, at the end of 2019, there were 1.32 million recreational personal drones in the United States [8]. In the statistical reports referred to in August 2021, there were about 869,428 drones registered in the United States by the Federal Aviation Administration (FAA), and 60% of registrations (517,974) were for entertainment purposes [9]. The Insider Intelligence website forecasts that sales of drones will top USD 12 billion by the end of 2021, and a large amount of that will come from the sale of personal drones used for photography, gaming, filmmaking, recording, and other entertainment activities [8]. Hobbyists will, however, spend USD 17 billion on developing various shapes of drones over the next few years. UAVs come in all sizes and models, from inexpensive small single-rotor devices to large, USD 1000+ quadcopters with GPS, multiple camera arrays, and first-person control. These shapes of drones are widely available, and the market is growing. As these technologies continue to grow and evolve, UAVs will become safer and more dependable in performing various tasks and will provide many services that were very difficult to perform and provide in the past.

The limitation of drones' batteries and the available optimal charging techniques represents a big challenge in using UAVs on a large scale. This problem makes UAVs unable to fly for a long time; hence, drones' performed tasks and provided services fail dramatically. Some requirements are necessary to guarantee the battery's best usage, which is one of the most important parts of flying a drone. When the drone is out of service, it is best to store the cool, dry battery. The battery is not left in the drone when it is out of service. This can help maintain the charge and increase the lifespan of the battery.

These battery charging requirements may increase the UAV's flight time, but this may be difficult to achieve due to various environmental conditions and constraints. Other requirements are with regard to the technical and engineering design of the battery itself. For example, the battery temperature increases after the flight and must be recharged when the flight battery temperature is below 40 degrees Celsius. Moreover, it is necessary not to overcharge the battery, which drastically decreases its lifespan. Fully-charged batteries cannot be fully stored for more than three days. If you do not let the electricity out, some

batteries will corrode, swell, and fail. Although some recent batteries may not swell, for the time being, the battery may be immediately damaged after several full charge saves. The Autel Robotics EVO Drone, which flies based on one lithium polymer battery, was the best in 2021, with the longest flight time of 25–30 min and a control range of 4300 m [10]. However, to enlarge UAVs' usage and benefits, it is mandatory to increase the flight time by more than only 30 min if we want to depend on UAV technology to perform critical activities in various domains and on a wider scale.

Due to these challenges, optimizing the scheduling of drone charging may be the unusual solution to drones' battery problems. Little work has been provided for addressing this problem. Most of the recent works focus on proposing drone schedule models for optimizing drones' routes for delivery services [11,12], monitoring services [13], or disaster inspection services [14].

In this paper, we address optimizing the scheduling of drone charging by investigating how to maximize the number of charging drones and minimize the number of dead drones based on the best configuration of drones and charging station points to increase the fight time of UAVs. To achieve this goal, we used two approaches. The first is the particle swarm optimization (PSO) algorithm [15]. It has been used to optimize drone routes toward ground station points to prevent collisions and congestion around charging points during charging flights. The second approach is the Stackelberg game theory [16] to optimize the scheduling of charging drones on charging station points based on charging factors such as mileage, battery status, arrival time, and charging price to achieve the best configuration of charging drones on charging station points. Another investigation is authenticating and securing the charging transactions between drones and charging stations using a novel blockchain protocol to prevent unauthorized charging transactions by intruder drones.

The proposed approach was simulated on 300 drones and 50 charging station points. The obtained results clarified the PSO algorithm's effectiveness for optimizing drone routes and preventing drone collisions during charging flights. Moreover, the proposed blockchain protocol has been simulated on Ethereum. The obtained results showed the effectiveness of blockchain in managing and securing charging transactions between drones and charging station points. In addition, the results showed the superiority of the proposed approach in scheduling drone charging compared to the proposed approach in [16].

This paper is organized as follows. Section 2 presents the literature review. Section 3 discusses the proposed approach. Section 4 presents the simulation results. Section 5 discusses the obtained results, and finally, Section 6 presents the conclusion of this study.

## 2. Literature Review

The existing literature on drone schedule models is extensive and focuses particularly on various civil services [17]. The recent attention has focused on introducing drone schedule techniques for shipment delivery and logistics services [18–20], monitoring services [13,21,22], and disaster inspection services [14,23]. However, a relatively small body of literature is specifically concerned with optimizing scheduling and securing drone charging problems using blockchain technology [24,25].

Over the past decade, most research in drone charging approaches has emphasized wireless power transfer (WPT) to enhance UAVs' charging [26–30]. Before the work of Hassija et al. in [16], optimizing the scheduling drone charging problem was neglected and had not been studied deeply.

Most drone routing and scheduling research have been carried out in various civil applications [31]. Some of these studies have assessed the efficacy of drone scheduling in logistics and shipment delivery. Pedro et al. [32] introduced a greedy heuristic technique to solve the truck–drone delivery problem based on multi-drop drone route planning. This technique is associated with a global optimization method using a simulated annealing (SA) algorithm to overcome routing problems such as the limited power-life of drone batteries and the identification of meeting points where fully-charged, new batteries replace them.

Moshref-Javadi et al. [33] proposed another metaheuristic approach for addressing the truck–drone routing problem based on one or multiple drones with synchronized movements and multi-drop drone routes. The authors defined the problem as a mixed-integer linear programming (MILP) model and performed a simulation study using the truck and drone routing algorithm (TDRA). The simulation results showed significant reductions in the customer waiting time based on the proposed multi-modal delivery model compared to a traditional truck-only delivery system.

Dyutimoy Nirupam Das et al. [34] addressed the truck–drone routing problem in logistics based on a multi-objective optimization model to minimize the travel costs and maximize the customer service level in terms of the suitable time of delivery. The practical results clarified that the proposed optimization model is an efficient method to parcel delivery logistics. An enhanced Pareto ant colony optimization algorithm is proposed and the non-dominated sorting genetic algorithm II (NSGA-II) is used to verify and validate the proposed mechanism.

Yong SikChang et al. [19] introduced another optimal delivery route based on a truck–drones approach. The main goal of this study was to find the wider drone delivery areas along a shorter truck route. After K-means clustering and traveling salesman problem modeling, the author solved this challenge by finding shift-weights with the non-linear programming (NLP) model after K-means clustering and traveling salesman problem (TSP) modeling. The practical results based on paired *t*-tests on randomly generated delivery locations showed the effectiveness of the proposed model compared with two other delivery route approaches

Without using a truck as a depot of UAVs in routing problems in logistics, Yan-chaoLiu [18] introduced a mixed-integer programming (MIP) model and proposed an optimization-driven, progressive algorithm for on-demand meal delivery using drones. The proposed algorithm is validated using simulation case studies, showing its efficiency in optimizing delivery operations.

Another drone routing and scheduling application is in monitoring and inspection [13,14]. The drone's routing strategy and scheduling algorithms need to be updated in real time to assist rescue crews in selecting the best ways to reach damaged areas. Jianfeng Fu et al. [35] proposed a real-time UAV routing methodology to arrange surveillance and inspection for post-disaster restoration in distribution networks. Based on this technique, real-time data about traffic conditions can be provided. The proposed performance has been validated in some case studies, and the results clarified the effectiveness of the proposed mechanism compared to other routing techniques in the literature.

Sudipta Chowdhury et al. [14] studied the post-disaster inspection drone routing problem using a mixed-integer linear programming model for minimizing the post-disaster inspection cost of a disaster-affected area based on several drone trajectory-specific variables. The authors proposed two heuristic algorithms called the Adaptive Large Neighborhood Search (ALNS), and the modified Backtracking Adaptive Threshold Accepting (MBATA) algorithm to solve the proposed optimization of the post-disaster inspection drone routing problem in a reasonable amount of time. The experimental results indicated the efficiency of the MBATA algorithm in solving the proposed optimization problem within a reasonable amount of time.

Waleed Ejaz et al. [24] proposed an energy-efficient task scheduling technique for data gathering by drones from IoT networks. The main objective was to optimize the path taken by drones to minimize power consumption. The important data collected by drones for people in disaster-affected areas were input into a decision tree classification algorithm to identify their health risk status. The risk status was then used to decide the areas that needed immediate assistance. The experimental results proved the effectiveness of the proposed scheduling technique with the traditional approach used for data collection and risk assessment in the literature.

Jun Xia et al., in [13], studied a drone scheduling problem (DSP) to monitor vessels in emission control areas. The DSP aims to design a set of flight tours for UAVs, including the

inspection procedures and timings for the vessels, so that as many vessels as possible can be verified during a given period while prioritizing highly weighted vessels for verification. To solve this problem, the authors proposed a Lagrangian relaxation-based algorithm that can produce near-optimal solutions for large-scale instances of the problem. The simulation results clarified that the proposed algorithm can effectively solve the DSP problem with near-optimal solutions.

Although the reviewed work introduced important and recent techniques to solve various drone routing and scheduling problems in many civil applications, little work has been provided to address the scheduling of drone charging problems within a network of drones and charging station points. Therefore, in this work, we study the optimization of the scheduling of drone charging problems. In addition, we investigate the efficiency of using blockchain technology to secure and authenticate the charging transactions between the scheduled drones and charging station points. Moreover, we secure and authenticate the charging transactions between UAVs and charging stations, especially the blockchain protocols.

## 3. Scheduling and Securing Drone Charging System

Traditionally, several approaches and techniques have investigated the drone battery charging problem to improve the efficiency of drone batteries to increase the drone flight time [30,36–38]. However, these techniques did not provide real solutions to ensure the longest flight period if the battery charge of the drones runs out. Therefore, developing an optimized solution to schedule drone charging on the charging station points is mandatory through a peer-to-peer network to ensure the drones' longest flying state. Another important issue that has been neglected by the related work is authenticating and securing the charging transactions between scheduled drones and charging station points. Few research efforts have been introduced to solve this problem, for instance, an auction-based approach to schedule drone battery charging using deep learning for multi-drone networks [39] and a consensus time-stamp and game theory-based system [16]. However, the drone charging scheduling problem is still in the early stages of research; key variables of the charging process such as drone location distribution and charging authentication and security have not been fully characterized in the literature. Therefore, this paper introduces a novel system for adding new contributions to scheduling and securing drone charging problems. The proposed system introduces two major operations as depicted in the architecture model in Figure 1. Registering drones' and charging stations' identities in the genesis block (i.e., the first block in the blockchain) is a pre-process for specifying the scheduling and verifying charging transaction requirements. The two major operations can be clarified as follows:

(1) *Charging scheduling:* using this operation, the proposed system utilizes three algorithms for optimizing drone scheduling on charging station points. The first algorithm is particle swarm optimization (PSO) [40], which maximizes drone routing to prevent collisions during charging flights. The second algorithm is proof-of-schedule (PoSch) consensus algorithm [41], which can be adapted for scheduling drone charging requests based on four random scheduling techniques. Finally, the third algorithm is the Stackelberg game-based auction [16] to optimize drone charging schedules produced by proof-of-schedule (PoSch) consensus algorithm

(2) *Charging authentication and verification*: using this operation, a proposed blockchain protocol can secure, authenticate, and verify the charging transactions between scheduled drones and charging station points. The main objective of this operation is to validate the drone charging requests to detect unauthorized drones in the blockchain network. The valid charging transactions are then encapsulated and securely stored in a new block, which is then added to the blockchain.

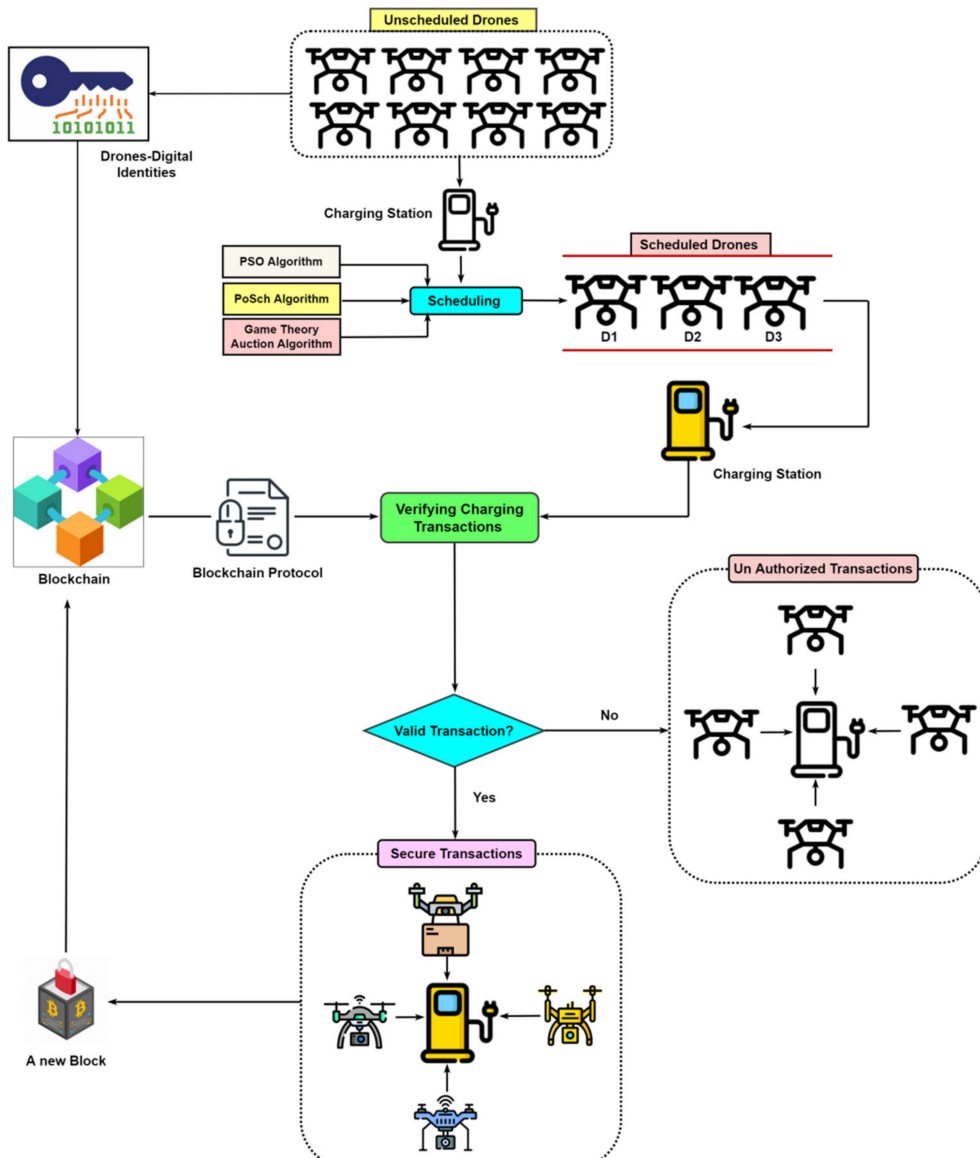

**Figure 1.** The architectural model of the proposed scheduling and securing drone charging system.

The methodology of the proposed system can be specified in Algorithm 1.

---

**Algorithm 1:** Scheduling and Authenticating Drone Charging Transactions

---

**Input** : $D = \{D1, D2, D3, \ldots \ldots . DN\}$
$\qquad S = \{S1. S2. S3, \ldots \ldots SM\}$
**Output** : Blockchain $= \{B1, B2, B3, \ldots \ldots . BN)$
Procedure: Scheduling and Authenticating Drone Charging Transactions
*While* $(D \text{ is } ! \text{ empty})$
$\quad X = \textbf{PSO} (\textbf{D}, \textbf{S}) \quad //$ optimization step using Particle Swarm Algorithm
$\quad Y = \textbf{PoSch} (\textbf{X}) \quad //$ Scheduling Step using Proof-of-schedule (PoSch) Algorithm
$\quad Z = \textbf{Verify} (\textbf{Y}) \quad //$ verifying charging transactions using a Blockchain Protocol
$\qquad \textbf{IF} (\textbf{Z} == 1)$
$\qquad\quad$ Blockchain= a new block
$\quad$ *Else*
$\qquad\quad$ **Print** ("Invalid Charging Transaction")
*End While*
*End Procedure*

---

*3.1. Authenticating Drones and Charging Stations' Identities*

Before explaining how drones are scheduled to charge at ground station points, it is mandatory to clarify how drones and ground stations are connected in a peer-to-peer network. What is the authentication technique used to add/remove a drone or a charging station to/from the network? The answer can be shaped based on two key techniques, ED25519 (EdDSA signature scheme using SHA-512 (SHA-2) and Elliptic Curve 25519) [42], and blockchain.

(1) **ED25519** is a public-key signature system based on Edwards-curve Digital Signature Algorithm (EdDSA). This recent and secure digital signature technique depends on the performance of optimized elliptic curves, such as the 255-bit curve (Curve 25519) [42]. Ed25519 uses short private keys (32 or 57 bytes), short public keys (32 or 57 bytes), and small signatures (64 or 114 bytes) with a high security level at the same time. The two generated ED25519-based keys (private and public) can authenticate the identities of drones and charge station points, as depicted in Figure 2. We use the ED25519 as an authentication technique due to the following attractive security features:

   (1) fast single-signature verification,
   (2) very fast message signing,
   (3) fast key generation,
   (4) small signatures and small keys,
   (5) high security level against side-channel attacks and twist-security attacks [43],
   (6) hash function collisions resilience,
   (7) foolproof session key.

(2) **Blockchain** is the second technique we use as a secure decentralized repository of the generated ED25519 public keys assigned to drones and charging stations. Moreover, blockchain can manage and verify the charging transactions between drones and secure charging stations. The applicability of using blockchain for storing transaction states in real time and verifying the validity of charging transactions can be justified based on a set of criteria explained in [44] and can be clarified as follows:

   (1) **Do you need to store the state of data?** The answer is YES because, in our proposed system, we need a decentralized repository (as blockchain) to store the generated ED25519 public keys assigned to drones and charging stations, which will be used later to verify charging transactions between drones and charging stations during a session of charging transaction. In addition, the details of the real-time transactions (i.e., states of a transaction between a drone and a station) have to be securely stored in this repository.

   (2) **Are there multiple participants?** The answer is YES because, in our system, drones and charging stations represent the nodes/participants that constitute a decentralized and P2P network system.

   (3) **Can you use an always online trusted third-party TTP?** The answer is NO because, in our system, a proposed blockchain consensus/protocol is responsible for verifying a charging transaction, which will be called automatically once a drone requests a connection with a charging station. Therefore, the blockchain protocol is the alternative verifier to TTP, which works only with a centralized system, but our drone–charging station system is decentralized. Hence. TTP is useless with our proposed approach.

   (4) **Are all participants known?** The answer is YES because, in our proposed system, we have a limited number of drones and charging stations labeled from 1 to N.

   (5) **Are all participants trusted?** The answer is NO because, in our proposed system, some drones or charging stations can work as fake or Sybille nodes; this normally requires a verifier to authenticate and validate charging transactions between drones and charging stations.

(6)   **Is public verifiability required?** The answer with simplicity is YES because transactions have to be validated before storing them in a new block in blockchain. Hence, we proposed a novel blockchain protocol to verify the drone charging process. Hence, our system can be classified as *public permission blockchain system*.

Figure 2 explains how drones and charging stations are authenticated based on ED25519 and blockchain techniques. The ED25519 key generator (e.g., SSH-keygen.) can generate public and private keys of drones and charging stations. Each drone/ground station holds its private key (PVK) and sends its public key to be stored in the blockchain, which will be used to verify charging transactions between drones and charging stations, as we explain in the following subsection.

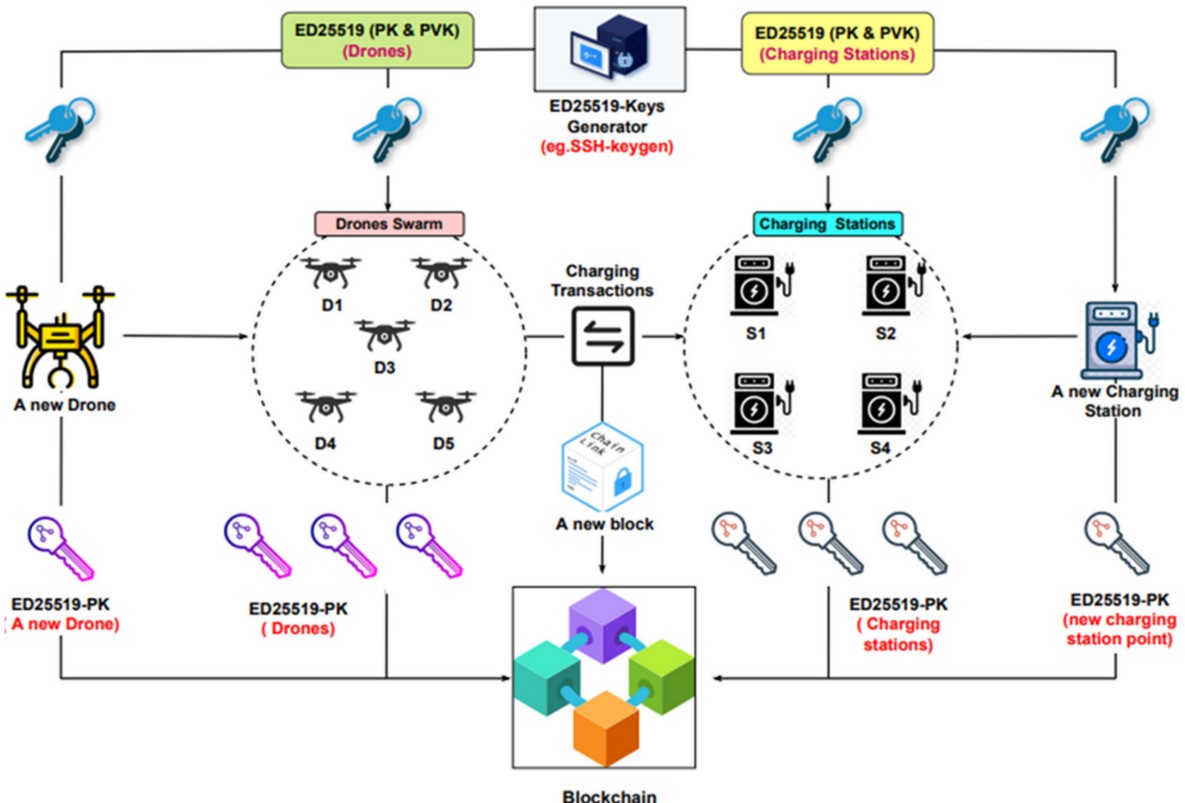

**Figure 2.** Authentication method of drones and charging stations.

### 3.2. Scheduling Drones' Charging Method

Before explaining how the scheduling drone charging problem can be solved, we have to clarify four principles:

(1)   All drones and charging stations are connected based on a P2P network, and the charging transactions are authenticated and verified using a blockchain protocol, as explained in Sections 4.1 and 4.2.

(2)   The PSO algorithm optimizes drone routes while flying toward charging stations to prevent collisions while requesting charging transactions.

(3)   The proof-of-schedule (PoSch) consensus algorithm [41] can be adapted to schedule drone charging requests on the charging stations in the blockchain network.

(4)   The Stackelberg game theory provided in [16] is then used to optimize the best scheduling technique that has to be followed by all drones. This is to maximize the number of drones that are allowed to charge and minimize the number of dead drones that do not have any available charging stations (dead drones here do not mean

crashed drones but mean they are a set of drones that cannot charge through this network; hence, these drones have to search for another network of charging stations).

The proposed scheduling drones' charging methodology can be described in the following steps:

***Step 1:*** *Optimizing drone routing using PSO algorithm*: The drone charging scheduling process should start through an optimized way to guarantee drones are collision-free while flying toward the charging station points. To achieve this goal, we used the methodology of the PSO algorithm, as depicted in Figure 3. For two reasons, we preferred the PSO algorithm over other optimization algorithms: (1) PSO has no overlapping and particle transformation calculation. This means the optimization methodology can be performed quickly based on the particle's speed. (2) The mathematical computation in PSO is very simple and easy to complete. It is hypothesized that utilizing the PSO algorithm requires drones and charging stations to be previously authenticated with the blockchain network, as explained in Section 3.1.

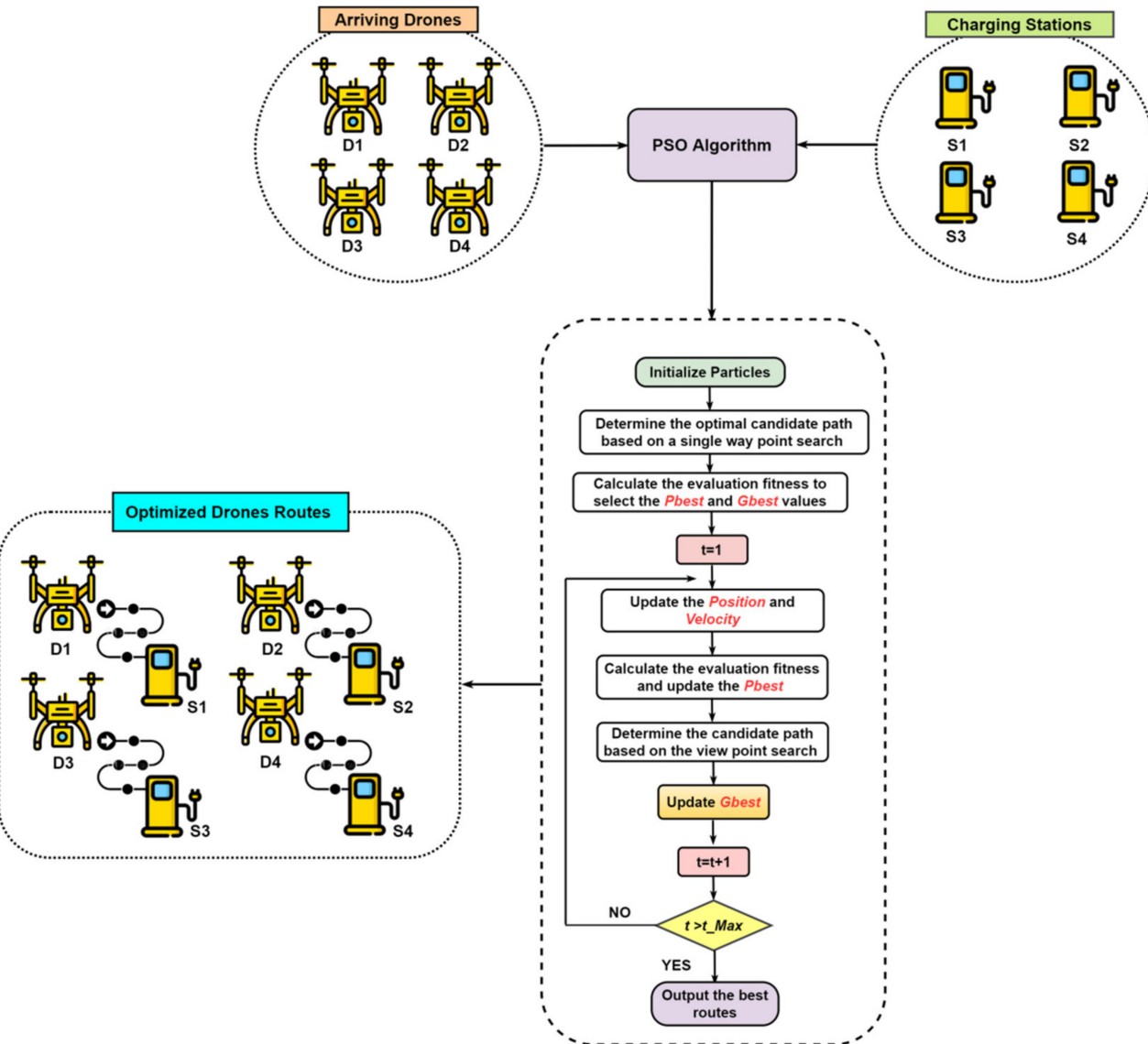

**Figure 3.** Optimizing drone routing using PSO algorithm.

***Step 2:*** *Scheduling drone charging requests using proof-of-schedule (PoSch) consensus algorithm:* PoSch [41] is a recently proposed consensus algorithm originally designed for task scheduling and resource allocation in cloud computing based on blockchain. The idea of

PoSch is to utilize the asymmetric Stackelberg game model to optimize scheduling clouds' pool of transactions. With our studied problem, scheduling drone charging requests, we can adapt the PoSch protocol to manipulate this problem but with different functions and features as follows:

(1) The new scheduling scenario is a drone–charging station not a client–cloud transaction.
(2) In scheduling drone charging problems, each charging station has two charging slots and can randomly select one of six scheduling algorithms of PoSch to schedule the arriving drones. These algorithms are the most common schedules used by the control processing unit (CPU), which deal with the issue of deciding which of the processes in the ready queue is to be allocated to the CPU: First-Come, First-Served (FCFS) scheduling, Shortest-Job-Next (SJN) scheduling, Priority (Pr) scheduling, Shortest Remaining Time (SRT), Round Robin (RR) scheduling, or Multiple-Level Queues (MLQ) scheduling [45]. In scheduling drone charging problems, these algorithms decide which drones are allocated to a specific charging station point in the ready queue. The successful drone (i.e., the first drone) is guaranteed a charge at the fixed, pre-set price of the charging station. It is allocated to the first charging slot in the charging station, while the other drones enter a game theory-based auction to decide the highest possible charging price to win the second slot in the charging station. The novel update of the PoSch methodology can be depicted in Figure 4. The PoSch protocol is randomly called three times with different scheduling algorithms (i.e., SRT, FCFS, and RR) for three different charging stations, S1, S2, and S3, respectively. In charging station S1, the drone D3 wins fixed price-based charging and is allocated to slot 1 of charging, while drones D1, D4, and D2 enter a game theory-based auction to decide which one will allocate slot 2. The same scenario is repeated in the charging stations S2 and S3 but with different schedules, as clarified in Figure 4.

***Step 3:*** *Optimal Drone Charging Schedules using a Stackelberg game model:*

To maximize the largest possible number of rechargeable drones (i.e., the best configuration of charging drones and charging stations and the least possible number of dead drones), the proposed Stackelberg game model provided in [16] can be imported here. In this step, the random schedules executed by the proof-of-schedule (PoSch) algorithm [41] in the previous step (as in Figure 4) decide which drone will enter the theory-based game auction and have to be optimized to guarantee the optimal drone charging schedules. The auction-based drones that request a charging transaction from a charging station must be authenticated in the blockchain network as explained in Section 3.1 and provide 5 attributes to become eligible for auction. These attributes include:

(1) the drone charging fingerprint code,
(2) the time-stamp of arriving drones to a charging station,
(3) mileage of the drone (i.e., the distance traveled by the drone per unit charge),
(4) drone battery status (i.e., how much drone's battery is available concerning the total capacity of the drone's battery),
(5) the price that the drone is ready to pay for being charged.

Figure 5 explains how drone charging schedules can be optimized based on the proposed Stackelberg game model [16]. The arriving drones are randomly scheduled using the proof-of-schedule (PoSch) algorithm based on the six scheduled routines depicted in Figure 4 (i.e., auctioned-based drones). Then, each drone has to provide the five charging parameters to the Stackelberg game algorithm, which provides the optimal drone charging schedules with the available charging station points. The dead drones produced from this process are then rerouted by the particle swarm optimization algorithm (PSO) to find the best routes to other charging station points and repeat the scheduling charging process.

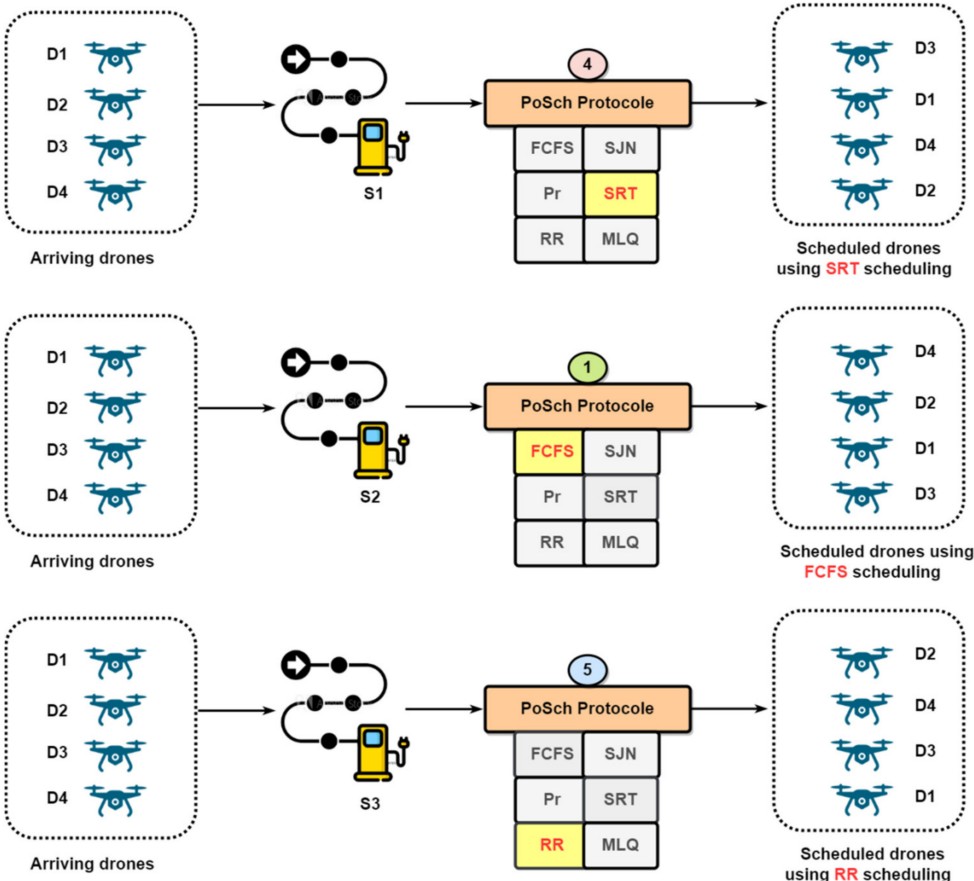

**Figure 4.** Scheduling drone charging requests using updating methodology of proof-of-schedule (PoSch).

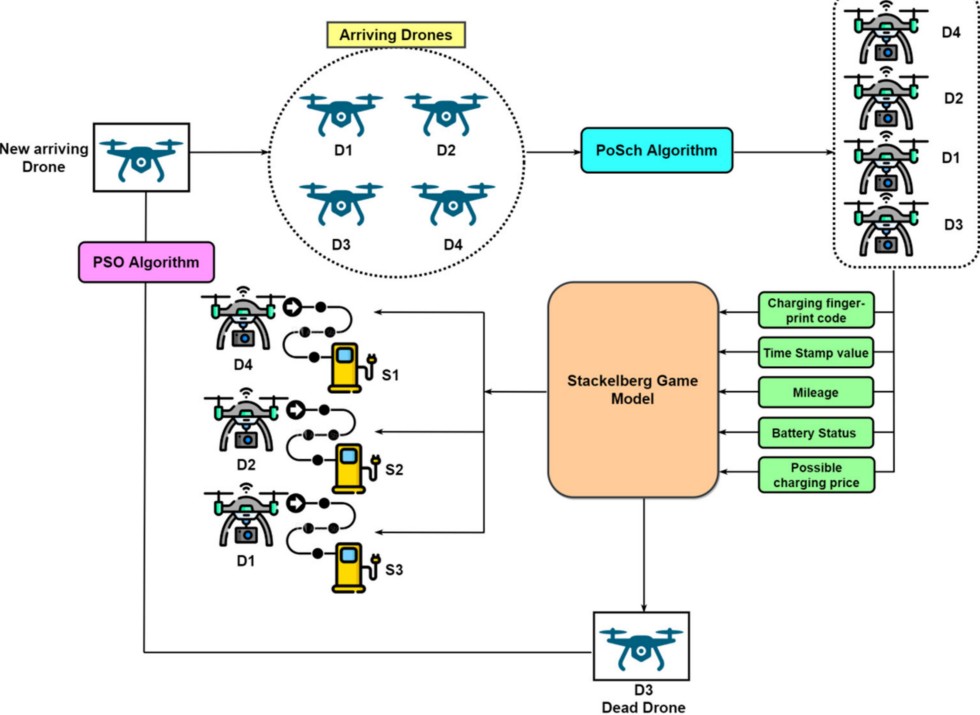

**Figure 5.** Optimal drone charging schedules model using the Stackelberg game algorithm.

### 3.3. Verifying Security of Charging Transactions

The validation of a charging transaction between a drone and a charging station point depends on a fingerprint code (called charging fingerprint) returned by blockchain that verifies that the private keys that drones and charging stations have match the corresponding public keys stored in the blockchain. The fingerprint value is calculated using an MD5 hash function, a built-in procedure in blockchain. Figure 6 explains how to verify charging transactions between a drone and a charging station point using a proposed blockchain protocol in 7 steps:

**Step 1:** A drone D sends a charging request to a charging station S.

**Step 2:** A charging station replies by asking for the private key (PVK) of the drone D.

**Step 3:** A drone D sends its PVK to the charging station S.

**Step 4:** Both of PVKs of the drone D and charging station S are then sent to block chain to verify their authenticity.

**Step 5:** Using the built-in MD5 hash function in the blockchain, a fingerprint code of the charging session is produced and returned to the drone, which requests a charging transaction. This can be achieved if both PVKs match the corresponding public keys of the drone D and charging station S stored in the blockchain.

**Step 6:** Drone D provides the fingerprint code of the session of charging to the charging station S.

**Step 7:** The drone D and charging station S check hands and establish a secure and valid charging transaction.

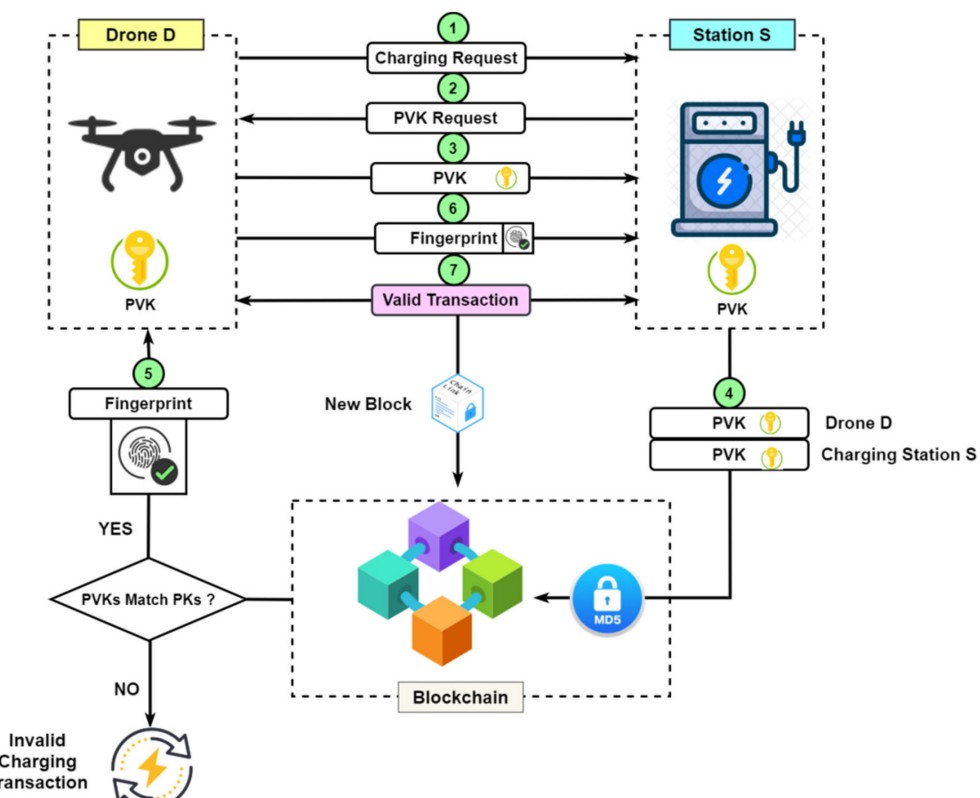

**Figure 6.** Verifying drone charging process using a proposed blockchain protocol.

### 4. Simulation and Experimental Results

To assess the performance of the proposed system, we conducted two simulation experiments on 300 drones and 50 charging stations. The first experiment was conducted to evaluate the optimization methodology of the proposed system in managing and scheduling drone swarms; this experiment was carried out on a machine with 2.3 GHz Intel Core 8

CPUs running Windows 10 Pro. Moreover, result visualization was performed on MATLAB R2019a, 64-bit version, and in Python programming language. The second experiment was conducted to evaluate the performance of the proposed blockchain protocol in managing and verifying charging transactions between drones and charging station points. This experiment was simulated on an Ethereum testnet and in Python.

*4.1. Drones' Optimization and Scheduling Results*

The particle swarm optimization (PSO) algorithm was applied on 300 drones and 50 charging station points to optimize the drone routes toward ground stations to prevent collisions and congestion around charging station points during charging flights. For an evaluation of how accurately the proposed model makes predictions for future observations of drone routes toward ground stations, many metrics were applied to measure the efficiency of PSO as an optimization method:

(1) Mean Absolute Error (MAE): It is a metric for evaluating error between two observations in the same experiment. Examples of Y versus X include comparisons of predicted values versus observed values. MAE is calculated as in Equation (1)

$$MAE = \frac{\sum_{i=1}^{N}|Y_i - X_i|}{N} \tag{1}$$

(2) Mean Square Error (MSE): This metric evaluates the absolute average distance between real and predicted values. MSE is calculated as in Equation (2)

$$MSE = \frac{\sum_{i=1}^{N}(Y_i - X_i)^2}{N} \tag{2}$$

(3) Root Mean Square Error (RMSE): is the standard deviation of the errors, which result when a prediction is made on a dataset. RMSE is the same as MSE, but the root of the value is considered while evaluating the accuracy of the model. MSE is calculated as in Equation (3)

$$RMSE = \sqrt{\frac{\sum_{i=1}^{N}(Y_i - X_i)^2}{N}} \tag{3}$$

where N is the total number of observations, $Y_i$ is the prediction value, and $X_i$ is the true value.

Figure 7 shows the PSO algorithm's MSE results on 300 drones and 50 charging stations across 200 iterations. After 10 iterations, the MSE curve drops steeply. This indicates that after a certain number of iterations, the PSO converges rapidly. Moreover, Figure 8 shows the MAE and RMSE results of the evaluating error rates between the real and prediction values. The results indicate the PSO achieved low error rates between the real values and prediction values regarding drone routes toward ground stations.

Drones arrive at a certain rate, and then they are directed using PSO to optimize their routes to avoid collisions and be normally distributed on chargers. Regarding drone scheduling, Figure 9 shows the losses of the drones that go without charging (i.e., dead drones) while applying the proposed scheduling methodology to 300 drones and 50 charging stations. The results indicate that the losses of dead drones are initially low with a low number of arriving drones, but as more drones arrive, the total number of dead drones uplifts. For more clarification, Figure 10 shows the percentages of the distribution of the losses of drones (i.e., dead drones) versus the simulation time (1400 min).

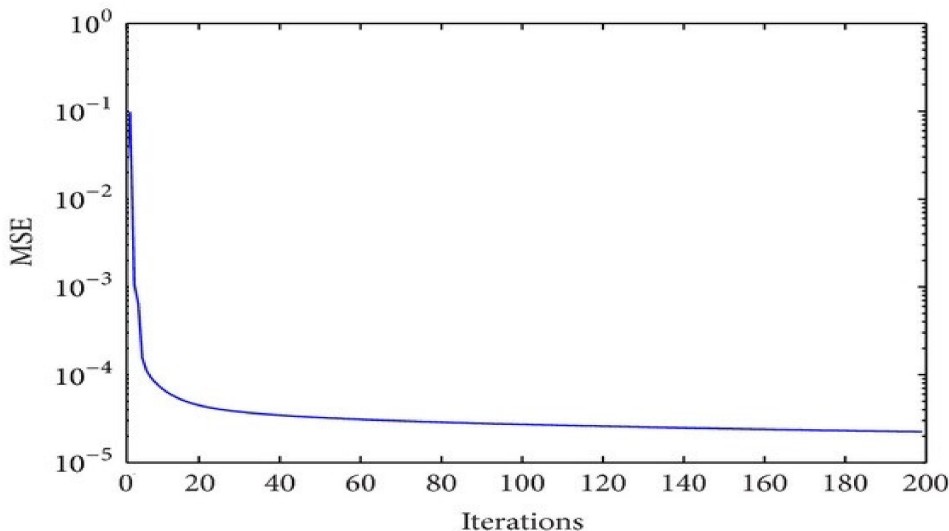

**Figure 7.** Results of Mean Square Error (MSE).

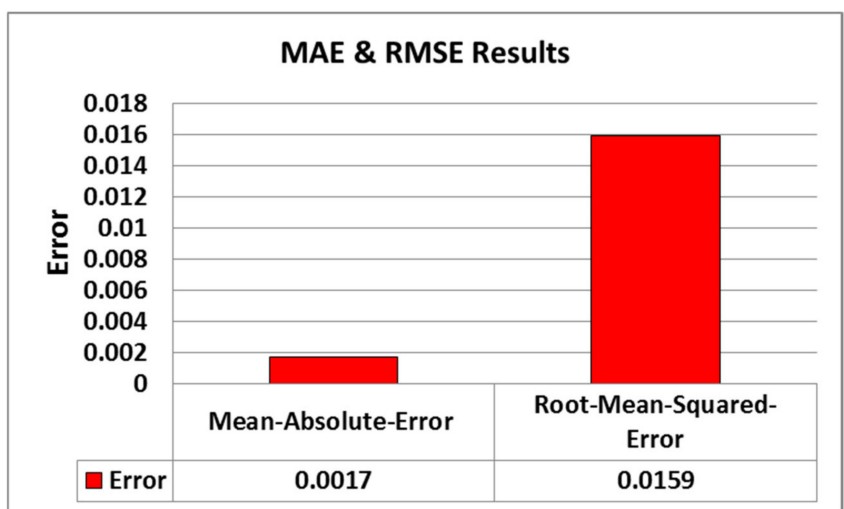

**Figure 8.** Results of Mean Absolute Error (MAE) and Root Mean Square Error (RMSE).

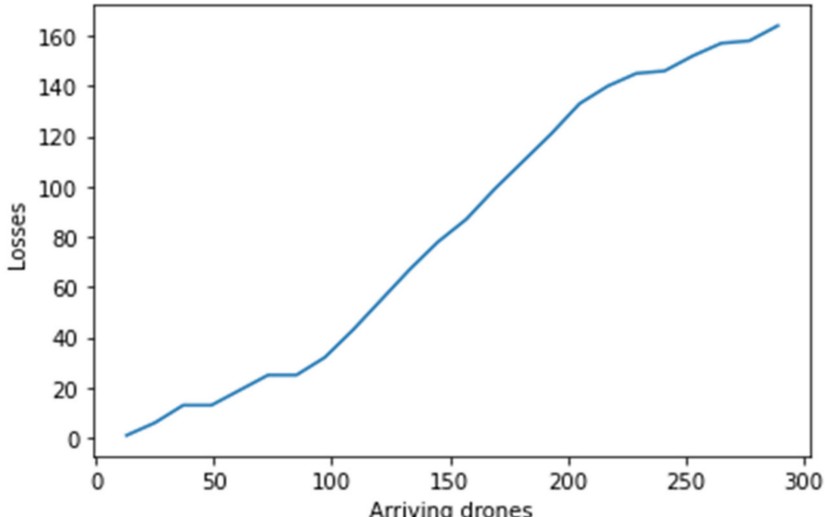

**Figure 9.** Drone losses (i.e., dead drones) vs. arriving drones.

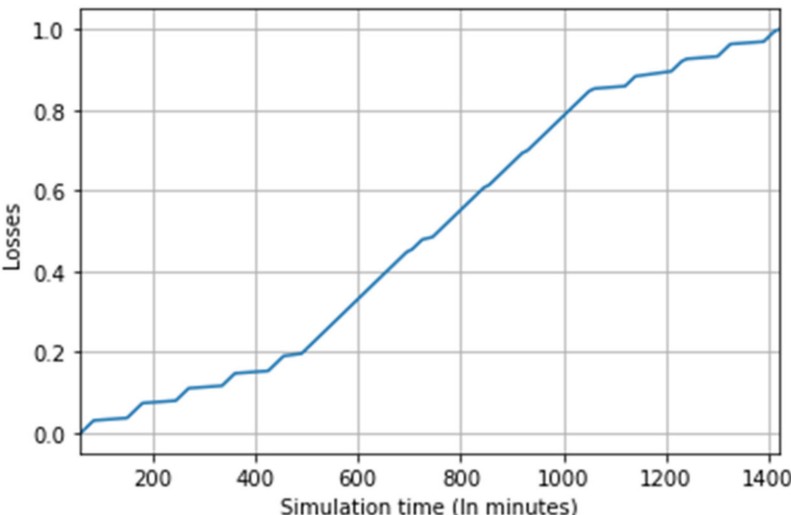

**Figure 10.** Drone losses (i.e., dead drones) distribution vs. simulation time.

*4.2. The Proposed Blockchain Protocol Results*

The proposed blockchain protocol (see Figure 6) was applied to 300 drones and 50 charging stations to evaluate its performance in verifying drones' charging transactions based on blockchain. Five metrics were used to evaluate the performance of the proposed blockchain-based protocol: read latency (RL), transaction latency (TL), read throughput (RTH), transaction throughput (TTh), and Ethereum GAZ.

(1)  *Read latency (RL).* Before defining read latency and it is computed, it is important to differentiate between reading and transaction operations in the blockchain. The read operation refers to an internal mechanism that a blockchain node can implicitly execute to fetch the required data to verify specific transactions. Still, it does not update the blockchain status. On the other hand, a transaction is a state transition that updates data in the blockchain by adding a new block to the chain. Therefore, a transaction is explicitly executed by a blockchain node and verified using a blockchain protocol against a set of rules called a smart contract. If a transaction is valid, the blockchain system will commit the transaction and add a new block to the chain containing all details of this transaction. Therefore, the RL is the time between the read request submitted by a drone and the charging response by the charging station. RL can be calculated as in Equation (4)

$$RL = Response\ Time - Submission \tag{4}$$

(2)  *Transaction latency (TL)* refers to a blockchain network-wide view of the amount of time taken for a transaction from creation to be available across the blockchain network according to the specified network threshold. TL can be calculated as in Equation (4)

$$TL = Confirmation\ Time\ - Submission\ Time \tag{5}$$

(3)  *Read throughput (RT)* is a measure of how many read operations are executed by drones in a specified time interval, where drones can perform several read operations per second (RPS) while they crest transactions with charging stations. Equation (6) is used to calculate RT

$$RT = \sum(Read\ Operations) / \sum(Times\ in\ seconds) \tag{6}$$

(4)  *Transaction throughput (TT)* is the rate at which the blockchain system executes valid transactions of drones with charging stations in a defined time interval, where many drones can do many transactions with many charging stations. This is not the rate

at a single blockchain node but across the entire blockchain network. This rate is expressed as transactions per second (TPS) at a specified network size. Equation (7) is used to calculate TT

$$TT = \sum (Committed\ Transactions) / \sum (Times\ in\ secondes) \tag{7}$$

(5) *Ethereum GAZ* refers to the fee required to successfully execute a transaction on the Ethereum blockchain platform. GAZ is represented as small fractions of the cryptocurrency ether (ETH), commonly called Gwei and nanoethics. GAZ is used to assign resources to the Ethereum Virtual Machine (EVM) so that decentralized applications, such as smart contracts can self-execute in a secured but decentralized manner. Hence, in our study, GAZ measures how computationally expensive a drone transaction is or how much charging transaction processing is needed.

The read latency and transaction latency results of simulating 300 drones on 50 charging stations are summarized in the Appendix A. Figure 11 depicts the averages of reading and transaction throughputs while simulating the proposed blockchain protocol. Moreover, Figure 12 depicts the numbers and percentages of the transaction on 14 blocks that contain all drones' charging transactions. Figure 13 depicts the Ethereum GAZ usage of 14 blocks.

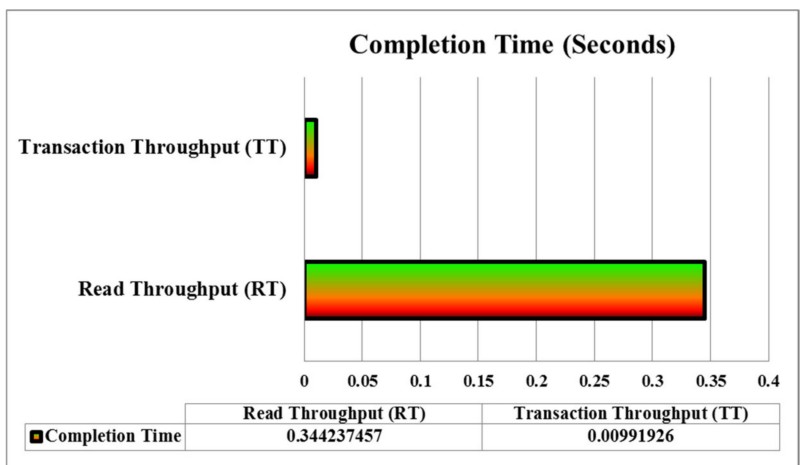

**Figure 11.** The completion time of drone charging results, average reading and transaction throughputs.

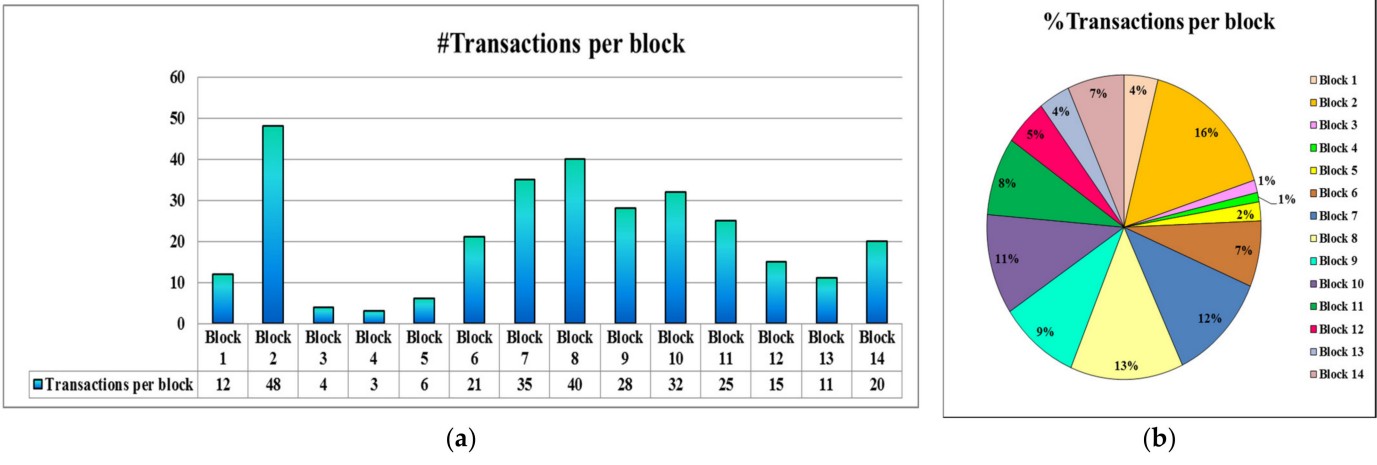

**Figure 12.** Transactions per block: (**a**) the number of transactions per block, (**b**) percentages of transactions per block.

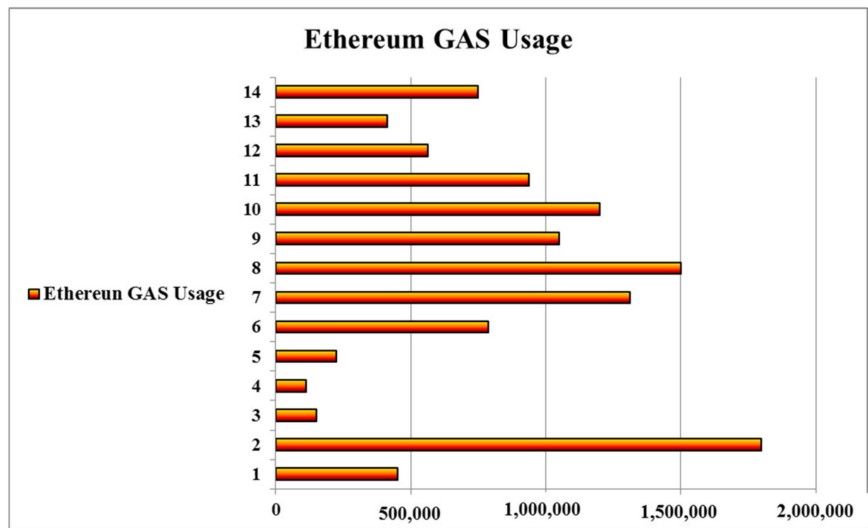

**Figure 13.** Ethereum GAZ usage of 14 blocks.

## 5. Discussion

Despite the growth rates of developing smart cities that utilize drone technology in various applications, very little was found in the literature on optimizing the scheduling of and authenticating drone charging. The recent work has focused only on introducing drone schedule techniques for shipment delivery and logistics services [18–20], monitoring services [13,21,22], or disaster inspection services [14,23–25]. Before the work [16], optimizing the scheduling of and authenticating drone charging was neglected and not studied deeply. In reviewing the literature, no mature approaches were found for the association between optimizing the scheduling of and authenticating drone charging transactions. Moreover, no work has been provided to discuss the efficiency of using blockchain technology in authenticating drone charging transactions with charging stations.

The first goal of this study sought to propose an efficient method for optimizing drone schedules on charging stations to maximize the number of charging drones and minimize the number of dead drones. In this study, the particle swarm optimization (PSO) algorithm provided good results for optimizing drone routes toward ground station points to prevent collisions and congestion around charging station points during charging flights. The simulation results of 300 drones and 50 charging stations showed that after 10 iterations of the PSO algorithm, the Mean Square Error (MSE) curve sharply decreased to $10^{-5}$. This indicates that the PSO achieved an efficient and accurate performance in optimizing drone routes with very low error rates, as depicted previously in Figures 7 and 8. Another important finding was shown when evaluating the efficiency of the model using the game theory-based auction algorithm provided in [16] to perform drone charging schedules to identify the number of dead drones that were lost without charging successfully compared to the successful ones (see Figures 9 and 10). Tables 1–3 summarize the simulation results of the numbers and percentages of the successfully charged drones and dead drones according to six classes of arriving drones and the associated number of charging stations through three rounds of the proposed scheduling methodology. Although the proposed scheduling methodology did not provide good results in the first round regarding the percentages of dead drones (i.e., drones could not be assigned to a charging station point), the recursive feature of the proposed scheduling method (see Figure 5) added a clear enhancement in the results. The proposed scheduling methodology achieved 96.8% of successful drone charging cases, while only 3.2% of drones failed to charge. The averages of the percentages of charging drones and dead drones over three rounds of drone charging scheduling of 300 drones and 50 charring station points are depicted in Figure 14.

**Table 1.** Drones' charging simulation results of **Round 1**.

| Round #1 | | | | | | | | | | | | |
|---|---|---|---|---|---|---|---|---|---|---|---|---|
| Arriving Drones | 50 | | 100 | | 150 | | 200 | | 250 | | 300 | | AVG |
| Number of Charging Stations | 8 | | 17 | | 26 | | 34 | | 42 | | 50 | | |
| Number and Percentages of Charging Drones | 41 | 82% | 78 | 78% | 83 | 56% | 98 | 49% | 117 | 46.8% | 136 | 45.4% | 59.5% |
| Number and Percentages of Dead Drones | 9 | 18% | 22 | 22% | 67 | 44% | 102 | 51% | 133 | 53.2% | 164 | 54.6% | 40.5% |

**Table 2.** Drones' charging simulation results of **Round 2**.

| Round #2 | | | | | | | | | | | | |
|---|---|---|---|---|---|---|---|---|---|---|---|---|
| Arriving Drones | 9 | | 22 | | 67 | | 102 | | 133 | | 164 | | AVG |
| Number of Charging Stations | 8 | | 17 | | 26 | | 34 | | 42 | | 50 | | |
| Number and Percentages of Charging Drones | 9 | 100% | 19 | 87% | 56 | 84% | 81 | 79.5% | 90 | 67.7% | 87 | 53.1% | 78.55% |
| Number and Percentages of Dead Drones | 0 | 0% | 3 | 13% | 11 | 16% | 21 | 20.5% | 43 | 32.3% | 77 | 46.9% | 21.45% |

**Table 3.** Drones' charging simulation results of **Round 3**.

| Round #3 | | | | | | | | | | | | |
|---|---|---|---|---|---|---|---|---|---|---|---|---|
| Arriving Drones | 0 | | 3 | | 11 | | 21 | | 43 | | 77 | | AVG |
| Number of Charging Stations | 8 | | 17 | | 26 | | 34 | | 42 | | 50 | | |
| Number and Percentages of Charging Drones | 0 | 0 | 3 | 100% | 11 | 100% | 21 | 100% | 43 | 100% | 64 | 84% | 96.8% |
| Number and Percentages of Dead Drones | 0 | 0% | 0 | 0% | 0 | 0% | 0 | 0% | 0 | 0% | 13 | 16% | 3.2% |

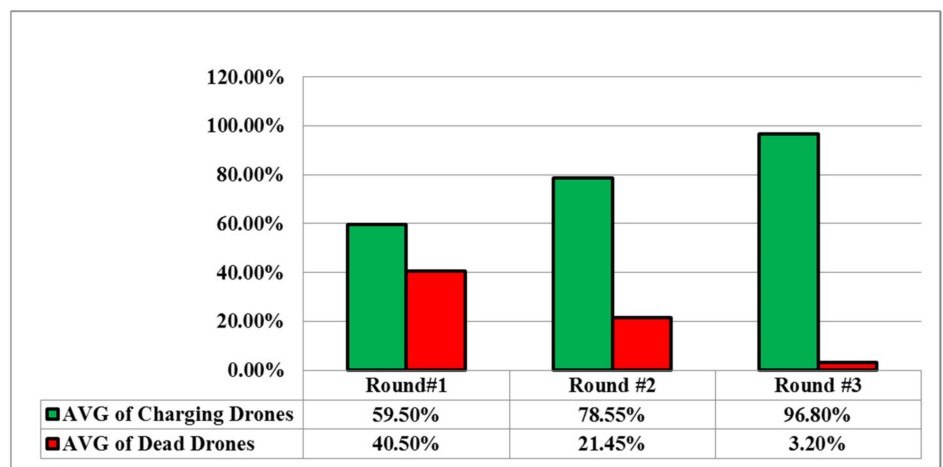

**Figure 14.** Averages of percentages of charging drones and dead drones over three rounds of drone charging schedule.

The second objective of this study was to investigate utilizing a novel blockchain protocol in authenticating and verifying drone charging transactions, as explained previously in Figures 2 and 6. The simulation results proved the good performance of the

proposed blockchain protocol in managing drones=charging station transactions within a short time and with low latency according to blockchain performance measures, read latency, transactions latency (refer to results in the Appendix A), read throughput, and transaction throughput (refer to Figure 11). These results mean that drones can conduct a successful and valid charging transaction rapidly within an average of 0.34 s.

Interestingly, these results are the first. We could not find a study in the literature introducing a mature solution based on conducting deep simulation experiments to optimize the scheduling of drone–charging station transactions. In addition, proposing and practically simulating a new blockchain protocol to authenticate and verify valid drone–charging station transactions is a novel contribution to UAV technology. It may grant this study an opportunity to be the base for future research studies in this domain.

This finding broadly supports the work [16] that investigated game theory to solve the scheduling drone charging problems. However, that study conducted a light simulation experiment on only four drones and three charging stations. Another issue is that the game theory-based auction model was not a deterministic technique to solve this complicated problem. The small numerical results based on light simulation showed only that the proposed technique provides a better price for the drones and better revenue for the charging stations based on evaluating energy rates. However, in our opinion, these results did not represent a real and mature solution for optimizing the scheduling drone charging problem.

The study also did not provide a real solution for manipulating the auction-out drones (i.e., dead drones). In addition, the authors mentioned that they used a security model based on a hash graph for securing drone transactions, but there are no numerical results for validating this proposal. These issues may make the current study better and mature enough to optimize the scheduling drone charging problems compared to [16]. Moreover, proposing and simulating the functionality of a novel blockchain protocol for authenticating and verifying drone charging transactions is superior in this study compared with [16] and other studies in the literature.

Although our results are promising, these findings may be somewhat limited by variables such as the number of drones, the number of charging station points, the optimization algorithm used to optimize drone routes, the hypothesis of drone batteries, charging flight conditions, and scheduling algorithms. These variables together mean that these results must be interpreted with caution.

Overall, the key results obtained from this study can be summarized in the following points:

- The investigation of optimizing the scheduling of drone charging has shown the efficiency of the PSO algorithm for optimizing drone routes and preventing the drones' collisions during charging flights with low error rates (MAE = 0.0017 and MSE = 0.0159).
- The proposed scheduling methodology based on the PoSch technique achieved 96.8% success in drone charging cases, while only 3.2% of drones failed to charge after three scheduling rounds.
- The simulation results in Ethereum proved the good performance of the proposed blockchain protocol in managing drone charging transactions within a short time and with low latency.

Finally, according to these results, we can infer that this study represents a real step for solving the scheduling of drone charging as an optimization problem. Moreover, introducing a novel blockchain protocol for authenticating and verifying drone charging transactions is a promising contribution to UAVs' security. The proposed protocol can also be adapted to electric vehicle charging technology.

## 6. Conclusions and Future Works

The present research aimed to provide a real solution for optimizing the scheduling drone charging problem. The second aim of this study was to investigate the efficiency of a novel blockchain protocol to authenticate and verify drone charging transactions. This study proposed a novel scheduling and secure drone charging system to achieve the

two goals. The proposed system was simulated on a generated dataset consisting of 300 drones and 50 charging station points to evaluate two major functions. The first function is optimizing the scheduling of drone charging using the particle swarm optimization (PSO) algorithm and game theory-based auction model. The second one is authenticating and verifying drone charging transactions using blockchain technology. The investigation into optimizing the scheduling of drone charging has shown the efficiency of the PSO algorithm for optimizing drone routes and preventing drones' collisions during charging flights with low error rates (MAE = 0.0017 and MSE = 0.0159). Moreover, the proposed scheduling methodology achieved 96.8% success in drone charging cases, while only 3.2% of drones failed to charge after three scheduling rounds.

The investigation into authenticating and verifying drone charging transactions showed the efficiency of the proposed blockchain protocol. The simulation results in Ethereum proved the good performance of the proposed blockchain protocol in managing drone charging transactions within a short time and with low latency. These results clarified that drones could rapidly perform a successful and valid charging transaction within an average of 0.34 s using the proposed blockchain protocol.

Before this study, evidence of optimizing the scheduling of and authenticating the drone charging problem was a purely anecdotal and incomplete solution. Therefore, these findings contribute in two directions to our understanding of optimizing scheduling drone charging and provide a basis for using blockchain technology to authenticate and verify drone charging transactions. The study should be repeated using other optimization and scheduling algorithms; moreover, applying blockchain attack models to the proposed blockchain protocol and evaluating its robustness would be a fruitful area for further work.

**Author Contributions:** Conceptualization: M.T. and M.E.-D. Data curation: M.T., M.E.-D. and E.G. Formal analysis, M.T. The investigation: M.T. and M.E.-D. Methodology: M.T., and M.E.-D. Senior Administration: A.E.H. Software: M.E.-D. and E.G. Supervision: A.E.H. and V.S. Validation: M.T. and M.E.-D. Visualization: M.T. Writing—original draft: M.T. Writing—review and editing, M.T., V.S. and A.E.H. All authors have read and agreed to the published version of the manuscript.

**Funding:** This research was funded with the support of the Slovak Operational Programme Integrated Infrastructure in the frame of the project: Intelligent systems for UAV real-time operation and data processing, code ITMS2014+: 313011V422 and co-financed by the European Regional Development Fund.

**Institutional Review Board Statement:** The study was conducted in accordance with the Declaration of Helsinki, and approved by the Institutional Review Board.

**Informed Consent Statement:** Informed consent was obtained from all subjects involved in the study.

**Data Availability Statement:** Exclude this statement since the study is supported by a created dataset.

**Conflicts of Interest:** Authors declare no conflict of interest.

## Appendix A. Read and Transaction Latency Results

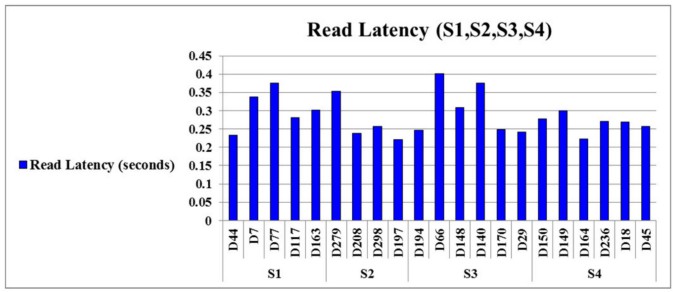 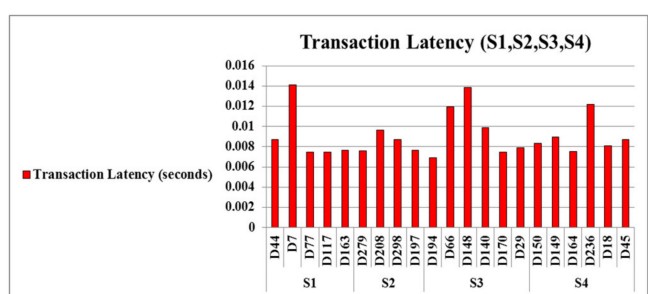

**Figure A1.** Read and transaction latency results of drones' charging transactions with stations S1–S4.

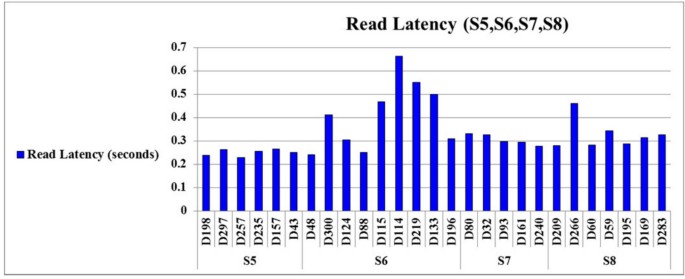
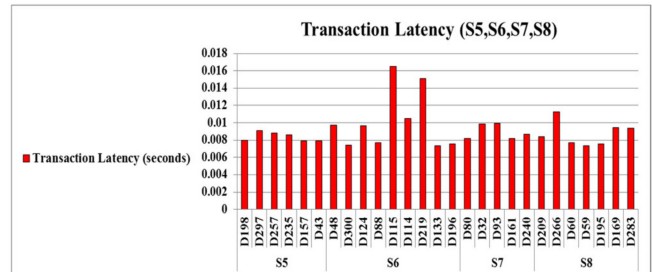

**Figure A2.** Read and transaction latency results of drones' charging transactions with stations S5–S8.

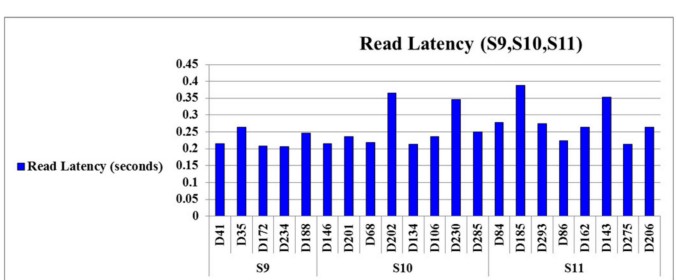
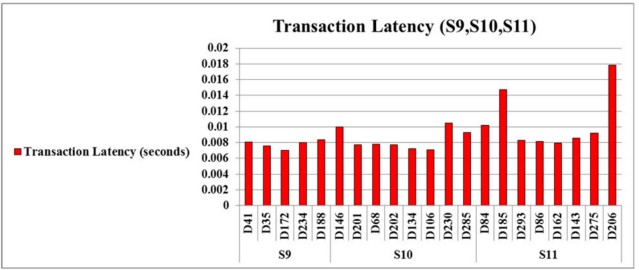

**Figure A3.** Read and transaction latency results of drones' charging transactions with stations S9–S11.

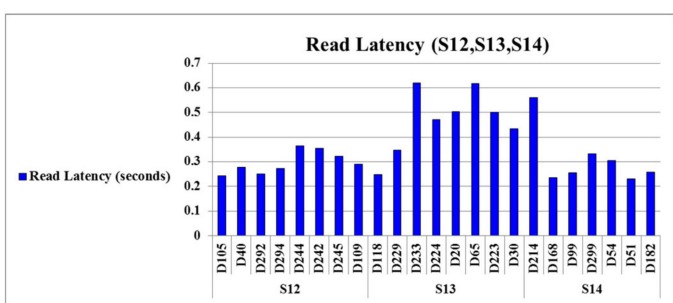
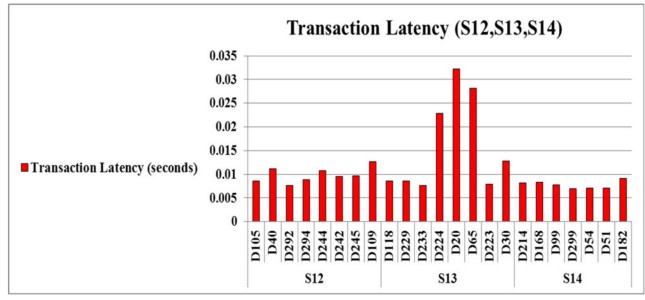

**Figure A4.** Read and transaction latency results of drones' charging transactions with stations S12–S14.

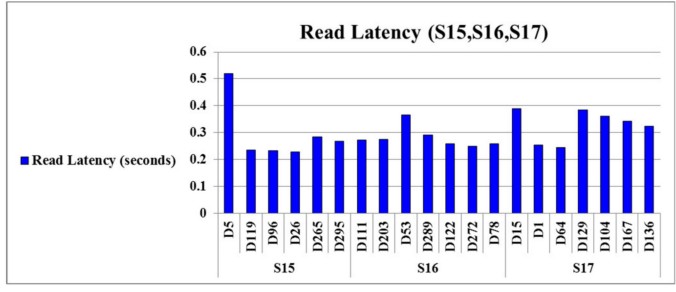
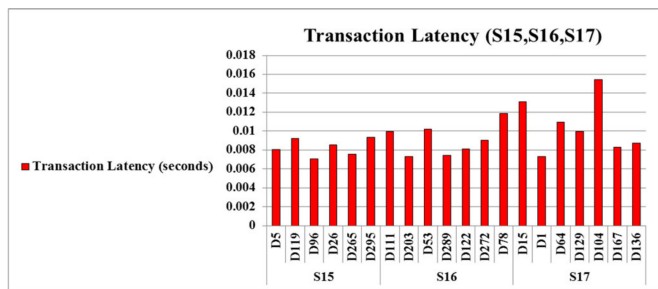

**Figure A5.** Read and transaction latency results of drones' charging transactions with stations S15–S17.

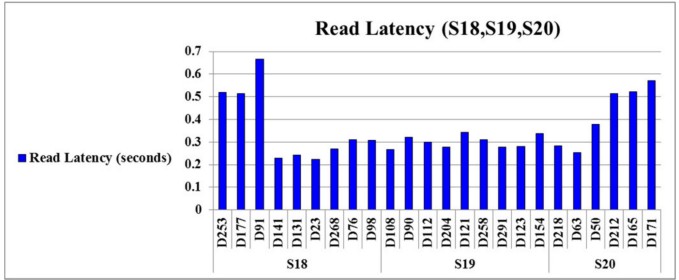
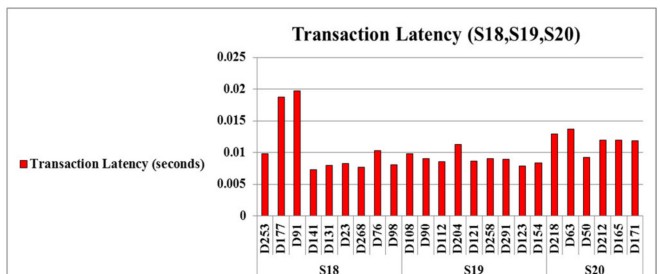

**Figure A6.** Read and transaction latency results of drones' charging transactions with stations S18–S20.

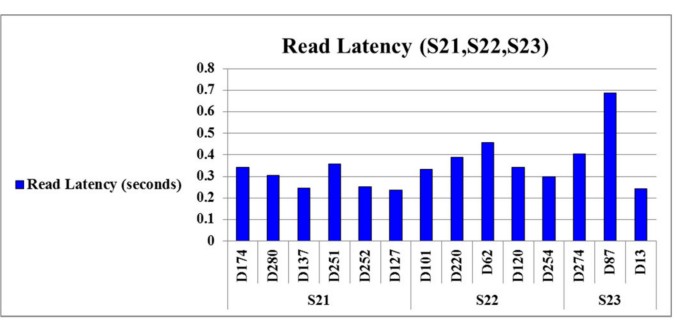
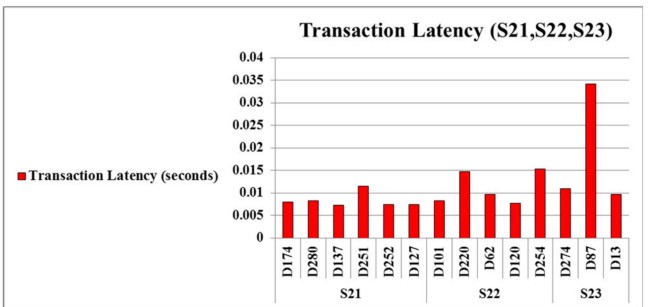

**Figure A7.** Read and transaction latency results of drones' charging transactions with stations S21–S23.

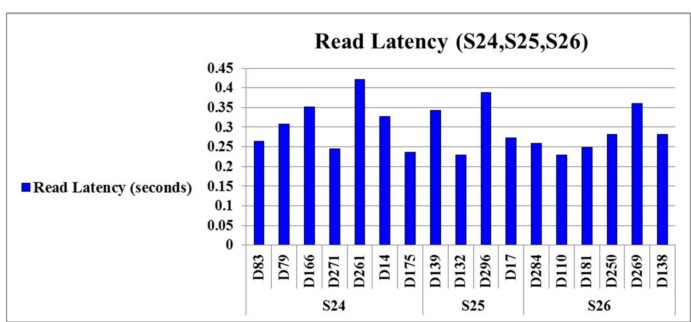
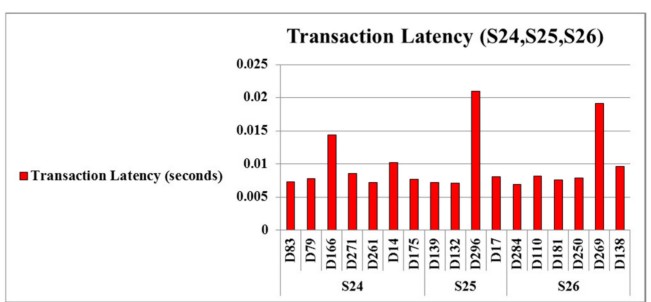

**Figure A8.** Read and transaction latency results of drones' charging transactions with stations S24–S26.

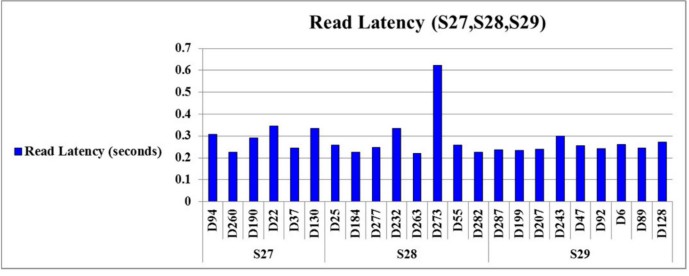
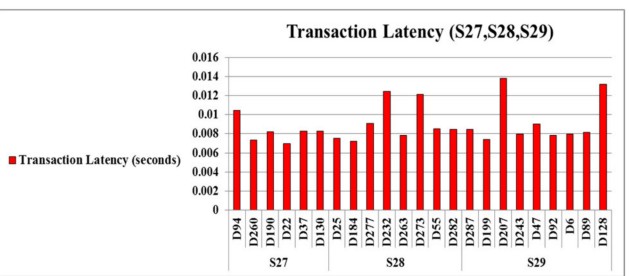

**Figure A9.** Read and transaction latency results of drones' charging transactions with stations S27–S29.

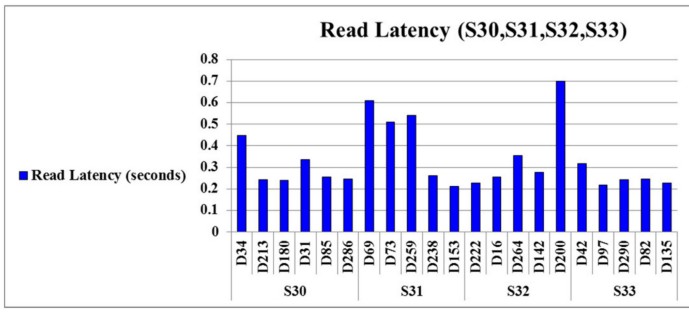
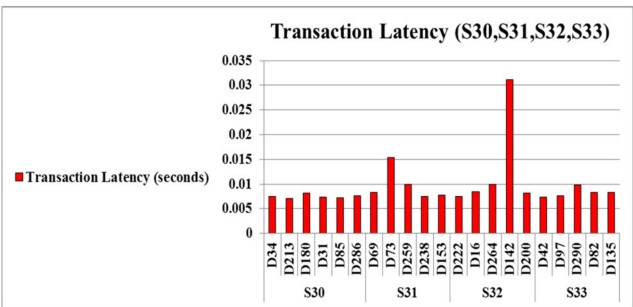

**Figure A10.** Read and transaction latency results of drones' charging transactions with stations S30–S33.

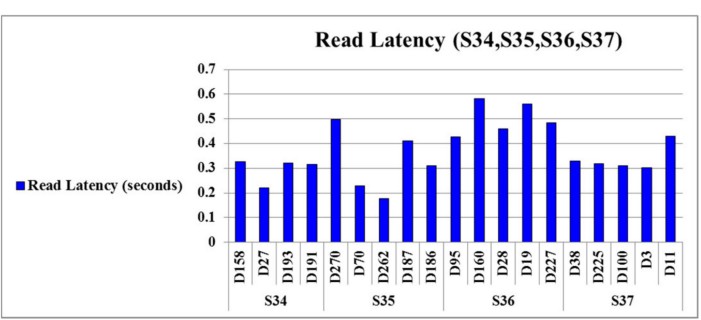
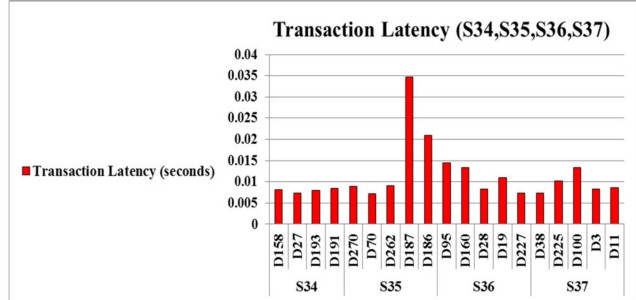

**Figure A11.** Read and transaction latency results of drones' charging transactions with stations S34–S37.

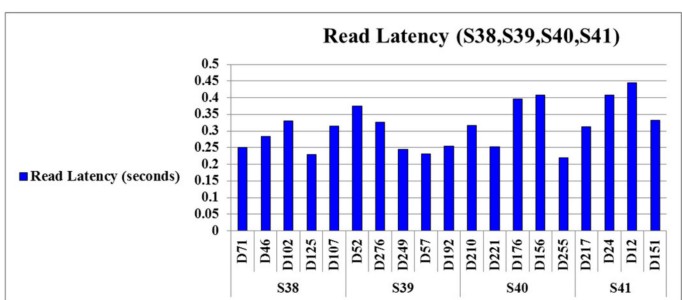
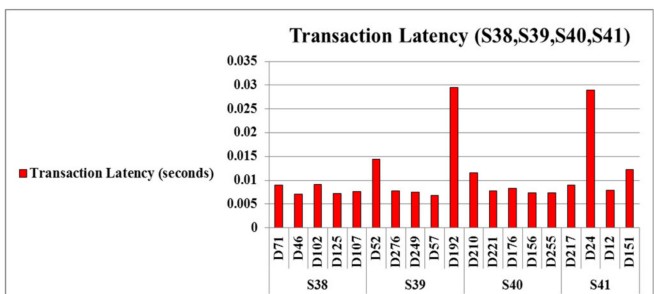

**Figure A12.** Read and transaction latency results of drones' charging transactions with stations S38–S41.

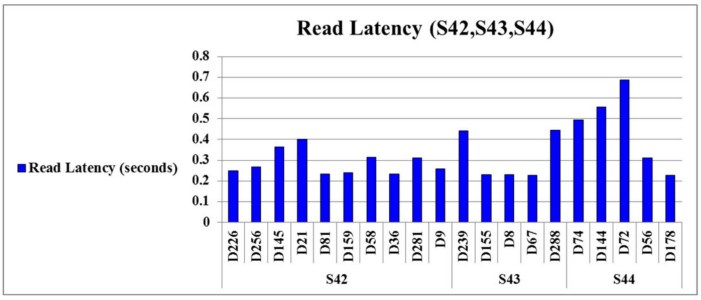
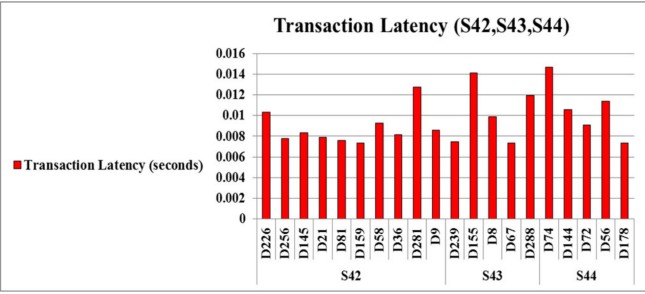

**Figure A13.** Read and transaction latency results of drones' charging transactions with stations S42–S44.

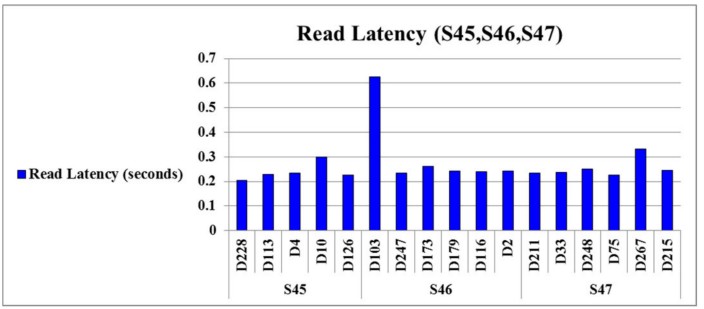
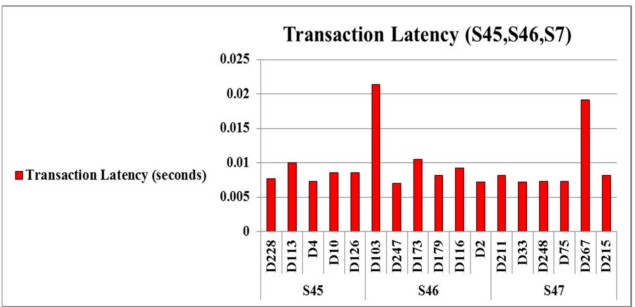

**Figure A14.** Read and transaction latency results of drones' charging transactions with stations S45–S47.

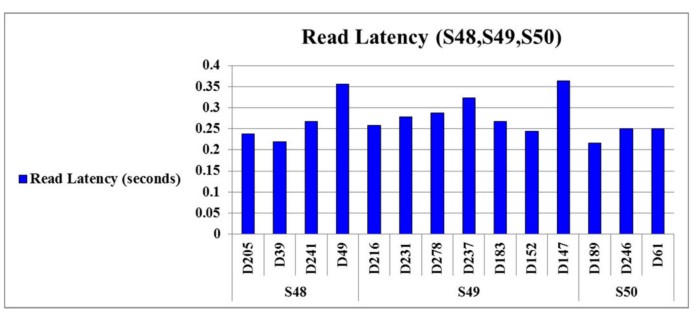
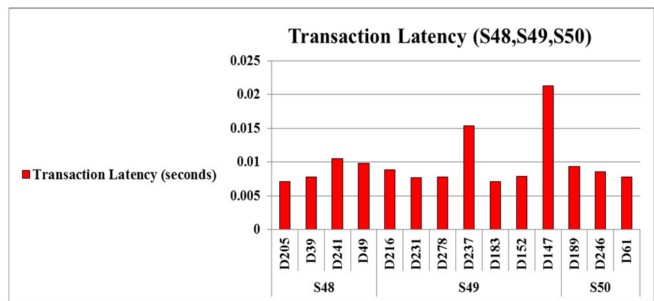

**Figure A15.** Read and transaction latency results of drones' charging transactions with stations S48–S50.

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
