# Peer review of "Scheduling and Securing Drone Charging System Using Particle Swarm Optimization and Blockchain Technology"

_drones, doi:10.3390/drones6090237_

Round 1
Reviewer 1 Report
I have just one simple comment to the authors. You need to consider the following reference publication regarding blockchain technology (ETH Zürich, Imperial College London):
Wüst, Karl, and Arthur Gervais. "Do you need a Blockchain?" IACR Cryptology ePrint Archive 2017 (2017): 375.
The default answer to the question is likely to be: no.
If you still think a blockchain is required please cite accordingly and thoroughly go through the presented decision criteria and explain.
Author Response
The point-by-point response is attached

Reviewer 2 Report
The purpose of the research presented in this article was to propose a solution to optimize the problem of charging drones at charging stations in order to maximize the number of charging drones.
The second objective of this study was to investigate the performance of the Blockchain protocol for authentication and verification of charging transactions.
The Particle Swarm Optimization (PSO) algorithm proved to be appropriate for optimizing drone routes, allowing to prevent collisions and congestion around charging points during charging flights. In addition, the results of the simulations showed good performance of the Blockchain protocol in managing drone charging transactions in a short time and with low latency according to Blockchain performance metrics.
Very interesting article, however, it seems too long and it is advisable to move some content (especially charts) to the appendix. I suggest moving to an appendix Figures 11-25 or presenting the results tabularly. Please standardize the bullet and caption styles of the figures (sometimes ending with a dot other times not).
After taking into account the above comments and the editing notes below, including organizing the content so that there is not so much empty space on the pages, the article can be published. It has the correct structure, the introduction and literature review do not raise any comments, except perhaps for one overly extensive citation ([26, 27, 28, 29, 30] in line 153). The illustrations are clear; I have indicated below only minor comments relating to them. The results need to be presented more concisely; "moreover" falls too many times in the discussion. The last chapter - summary and directions for further research also does not raise comments.
During content analysis, the following editing errors were noted, it is advisable to correct them before publication:
special purpose UAVs in military and civil operations. [6]. It will be great to develop new 72->
special purpose UAVs in military and civil operations [6]. It will be great to develop new 72
line 456-458:
Step 6: Drone D provides the fingerprint code of the charging session to the charg-
ing
station S
3. ? Mean Square Error (RMSE): is the standard deviation of the errors which result 507
Figure 7. Mean Squared Error(MSE) results->
Figure 7. Mean Squared Error (MSE) results
Figure 8.-> It would be consistent to post the figure without the border and title above
Figure 10. Drones losses (i.e. dead drones) distribution Vs. Simulation Time->
Figure 10. Drones losses (i.e. dead drones) distribution vs. Simulation Time
drones 'charging transac-564->
drones' charging transac-564
Line 620:
Figure 28 depicts Ethereum GAS usage of 14 blocks
Figure 28 depicts Ethereum GAS usage of 14 blocks.
Figures 11-29.-> It would be consistent to post the figure without the border and title above
Rotate the legends on the vertical axis by 90 degrees, making the charts slightly larger
Figure 26. the completion time of drone->
Figure 26. The completion time of drone
Figure 27. transactions per block: a) #Transactions per block, b) %Transactions per block->
Figure 27. Transactions per block: a) numbers of the transaction, b) percentages of the transaction
Tables 1-3: "#&%" ?
Figure 29. not 0,00%... just 0, 20, 40... 100%
Finally, According to these results, we can infer that this study represents a real step for 743->
Finally, according to these results, we can infer that this study represents a real step for 743
Author Response
The Point-by-Pint response file is attached

Reviewer 3 Report
The paper proposed a scheduling and securing drone-charging system using Particle Swarm Optimization algorithm. Some major corrections are required:
1. Is it system model or network architecture? section 3. Scheduling and Securing Drone Charging System
2. Please give some mathematical model in your system.
3. In eq. (5), Confirmation Time@Network Threshold ?? is it Confirmation Time x Network Threshold
4. In eq. (6), what is the meaning of sum symbol. Is it all UAV or another things?? Same to eq. (7)
5. Please give the key points of your results before 6. Conclusion and future works
Author Response
The Point-by-Point response file is attached

Round 2
Reviewer 3 Report
Thanks for correction.